# Efficacy of Fractional CO_2_ Laser Treatment for Genitourinary Syndrome of Menopause in Short-Term Evaluation—Preliminary Study

**DOI:** 10.3390/biomedicines11051304

**Published:** 2023-04-28

**Authors:** Andrzej Woźniak, Sławomir Woźniak, Ewa Poleszak, Tomasz Kluz, Łukasz Zapała, Aleksander Woźniak, Tomasz Rechberger, Andrzej Wróbel

**Affiliations:** 1Independent Public Clinical Hospital No. 4, 8 Jaczewskiego, 20-954 Lublin, Poland; 2Third Department of Gynecology, Medical University of Lublin, 8 Jaczewskiego, 20-090 Lublin, Poland; 3Laboratory of Preclinical Testing, Chair and Department of Applied and Social Pharmacy, Medical University of Lublin, 1 Chodźki Street, 20-093 Lublin, Poland; 4Department of Gynecology, Gynecology Oncology and Obstetrics, Institute of Medical Sciences, Medical College of Rzeszow University, 16c Rejtana, 35-959 Rzeszow, Poland; 5Clinic of General, Oncological and Functional Urology, Medical University of Warsaw, 4 Lindleya, 02-005 Warsaw, Poland; 6University Clinical Center of the Medical University of Warsaw, 1A Banacha, 02-097 Warsaw, Poland; 7Second Department of Gynecology, Medical University of Lublin, 8 Jaczewskiego, 20-090 Lublin, Poland

**Keywords:** GSM, menopause, atrophy, laser CO_2_

## Abstract

The postmenopausal state covers 40% of modern women’s lives and 50–70% of postmenopausal women report GSM symptoms such as vaginal dryness, itching, frequent inflammations, lack of elasticity, or dyspareunia. Consequently, a safe and effective method of treatment is crucial. In a group of 125 patients, a prospective observational study was performed. The aim was to evaluate the clinical effectiveness of fractional CO_2_ laser in the treatment of GSM symptoms using a protocol of three procedures in 6-week intervals. The vaginal pH, VHIS, VMI, FSFI, and treatment satisfaction questionnaire were used. The fractional CO_2_ laser treatment was effective in improving all the objective forms of evaluation: vaginal pH (from 5.61 ± 0.50 at the baseline up to 4.69 ± 0.21 in the 6-week follow-up after the third procedure); VHIS (12.02 ± 1.89 at the baseline vs. 21.50 ± 1.76); VMI (21.5 ± 5.66 vs. 48.4 ± 4.46). Similar results were obtained for FSFI: 12.79 ± 5.351 vs. 24.39 ± 2.733, where 79.77% of patients were highly satisfied. Fractional CO_2_ laser therapy increases the quality of life by having a beneficial effect on the sexual function of women with GSM symptoms. This effect is obtained by restoring the correct structure and proportions of the cellular composition of the vaginal epithelium. This positive effect was confirmed by both objective and subjective forms of evaluating GSM symptom severity.

## 1. Introduction

Genitourinary syndrome of menopause (GSM) is a serious and common issue for many women. Menopause occurs in most women around the age of 51 ± 4 years (45–55 years of age) [1]. According to the latest reports, the postmenopausal state covers 40% of modern women’s lives and 50–70% of postmenopausal women report GSM symptoms [2]. According to WHO reports, it is estimated that in the year 2030, there will be up to 1.2 billion women of postmenopausal age living around the globe [3]. These data show the scale of this phenomenon and represent the magnitude of the issue that modern women as well as modern medicine has to face.

The gradual decline of the hormonal activity of the ovaries during menopause has a significant impact on a woman’s body. The most common symptoms are hot flashes, mood swings, episodes of depression, osteoporosis, and metabolic and cardiovascular changes [4]. However, the decrease in the amount of estrogen results primarily in numerous changes in the female genitourinary system. The progressive decrease in the level of estrogen in females leads to a decrease in the activity of fibroblasts in the vaginal mucosa, which results in a reduction in the amount of collagen, which maintains the specific support for the epithelium and ensures the correct biocenosis of the vagina. The clinical effect of this phenomenon is smoothing of the vaginal folds and thinning of the epithelium. Additionally, the process leads to disturbances in the statics of the pelvic organs and symptoms of urinary incontinence, associated with a reduction in the quality of connective tissue and the thinning of the vaginal wall muscles. Vaginal dryness, irritation, and dyspareunia associated with these conditions appear most often in the early postmenopausal period, and sometimes even in the months preceding the onset of the last menstruation. Such symptoms cause a decrease in the quality of sexual life, in extreme cases leading to complete resignation from this aspect of life [5,6,7,8].

The symptoms of GSM are very burdensome, making daily and professional activities difficult, as well as significantly affecting the mental and emotional sphere of women. Symptoms of urogenital atrophy (UA) such as vaginal atrophy (VA) are often accompanied by symptoms of overactive bladder (OAB) [9,10]. 

In the postmenopausal period, epithelial thinning, as a result of a reduction in the number of glycogen-rich cells in the superficial layer, leads to changes in the vaginal flora, and in particular to a reduction in the lactobacilli population. As a consequence, it increases the pH level to the alkaline direction and thus promotes changes in the microflora in favor of pathogenic bacteria. As a result, this leads to the occurrence of symptoms typical for UA in the form of vaginal inflammation, itching, discomfort, and irritation of the vaginal mucosa. The appropriate pH level is crucial for maintaining the proper bacteriological condition of the vagina [11,12].

The process of gradual disappearance of the stratified squamous epithelium lining the vaginal mucosa during menopause is associated with changes in the proportions between its cell types. The vaginal maturation index (VMI) is used to assess these proportions [1,13]. It allows visualizing the percentage ratio between the increasing number of the intermediate and parabasal cells compared to the cells of the superficial layer [14]. For menstruating women, the VMI range is considered to be between 50 and 100. Values between 65 and 100 indicate high levels of epithelial estrogen stimulation. In postmenopausal women with symptoms of UA where estrogen stimulation is insufficient, VMI values are <50 and decrease with increasing severity of GSM [15,16,17].

Although the measurement of vaginal pH and VMI is a standard criterion for confirming the presence of UA, it is generally not used in clinical practice due to the fairly clear nature of GSM symptoms. 

The standard of treatment is estrogen therapy, but there are large groups of patients in which it is contraindicated, or in which it only brings negligible effects. The main limitation of estrogen therapy is its contraindication in a group of patients with estrogen-dependent cancer history. In the case of vaginal treatment, there is a lack of a systematic approach on the part of the patients [16,17,18].

Recently, more and more attention has been paid to the possibilities of using energy-based devices in urogynecology, especially CO_2_ fractional lasers. This resulted in a significant increase in the number of therapies with their use in the treatment of GSM including static disorders of the pelvic organs, urinary incontinence, as well as UA in general [19].

Laser therapy is a new, non-hormonal method of treatment for UA symptoms. The principle of laser procedures in the treatment of VA states is to use a wavelength with high water absorption, such as a carbon dioxide laser, to ablate and coagulate the vaginal or vulvar tissues. The laser’s mechanism principle is based on a direct effect on the vaginal mucosa, which induces the metabolic activation of fibroblasts. The repair reaction initiated by the thermal effect of the laser beam leads to the remodeling of vaginal tissue with the neoformation of collagen fibers and elastin in the atrophic epithelium. The laser beam, by affecting the epithelial cells, releases its energy, which stimulates the epithelium to mature and results in its thickening (increase in VMI) and exfoliation. Moreover, an increase in the formation of papillae present in the epithelium and an increase in angiogenesis of the network of vessels that supply has been shown, which ensures better blood supply to the epithelium, and thus results in an improved supply of nutrients [2,12,20].

## 2. Materials and Methods

The primary study group consisted of 159 women in the postmenopausal period. The inclusion criteria were menopausal status, one or more GSM-related symptoms, and informed consent to participate in the study by the patient. The exclusion criteria were abnormal Pap smears, recurrent urinary tract infections or active genital infection, systemic steroid or hormonal use in the previous three months, pelvic organ prolapse (POP) > II, or lack of informed consent. This prospective study initially included a population of 125 patients aged 41 to 69 who reported symptoms of GSM. After the 1st laser treatment, only 89 patients decided to undergo the 2nd treatment. All of the 36 patients who left the treatment at this stage were satisfied with only one laser procedure. The whole laser therapy was finished by 84 patients, who were then evaluated as the study group (Figure 1). 

The aim of the experiment was to evaluate the clinical effectiveness of the use of CO_2_ laser in the treatment of GSM symptoms in postmenopausal patients, taking into account its impact on short-term therapeutic efficiency (up to 6 weeks after the last laser treatment). The whole study was approved by the local ethics committee (approval no. KE-0254/127/2019).

According to the experiment scheme, each patient qualified for the therapy had to undergo 3 treatments with a Monalisa Touch CO_2_ laser by DEKA at 6-week equal intervals [21,22]. On every visit, the patient underwent a gynecological examination, during which the vaginal pH was measured and the severity of UA was assessed using the vaginal health index score (VHIS). Consequently, vaginal elasticity, pH, fluid secretion, epithelial mucus membrane, and moisture were assessed in the range of 1–5 points, with a total score of 5–25 points. Then, the cytological smear was taken from the upper third of the vaginal wall and then was evaluated by light microscopy to determine VMI. For this purpose, 100 cells present in the field of view were counted in fixed and stained cell smear preparations, dividing them into superficial, intermediate, and basal cells. We used 5×, 10×, and 20× magnification for correct cell identification. The Multi Scan v5.10 system was used to correctly determine the number of cells. Then the patient completed the following FSFI questionnaire. An additional questionnaire to be completed by the patients was the “Satisfaction with the procedure questionnaire”, in which patients were asked to respond on the visit after 6 weeks from the last treatment. The laser manufacturer recommends 3 treatments at 30–40 days intervals and then the level of satisfaction with the full procedure should be evaluated.

## 3. Results

Before the therapy, the pH of the vaginal environment in the studied group of patients was 5.61 on average and ranged from 4.70 to 7.00 (Table 1). After each laser treatment, the pH level decreased significantly in relation to the condition before the treatment as well as in relation to each of the previous treatments. The lowest pH level was achieved 6 weeks after the third treatment and averaged 4.69, ranging from 4.40 to 5.30 (Figure 2).

The overall assessment of VHIS of patients before the therapy was on average 12.02 and ranged from 7.00 to 16.00. The lowest result measured on the VHIS scale was determined for the parameters epithelial mucous membrane (2.15) and vaginal secretions (2.17), and then for the parameters pH (2.39), vaginal hydration (2.60), and vaginal elasticity (2.70). Before the first treatment, of 125 patients, 4.0% had an overall VHIS score slightly above the cut-off value of 15.

After only the first treatment, the proportion of patients whose overall VHIS exceeded 15 increased to 71.9%, and of the 125 patients tested, only 28.1% had an overall score of ≤15 (Figure 3).

A total of 84 patients underwent the whole set of laser three treatments. Six weeks after the last (third) laser treatment, the average VHIS for this group of women was 21.50 and ranged from 17.00 to 25.00. There was a significant improvement in all parameters of VHIS compared to the state before the procedure, and also in relation to the values obtained after the first and after the second procedure. The lowest value of the VHIS was obtained for the parameter vaginal secretion (3.92), and then for the following parameters: vaginal hydration (4.05), pH (4.43), epithelial mucous membrane (4.45), and the highest score, as with the previous treatments, was noted for the elasticity (4.66). The overall vaginal health index score also increased significantly, which in the entire study population exceeded the value of 15 (Table 2).

The pretreatment VMI of the patients was on average 21.5% and ranged from 11.5 to 32.0. Cells of the basal layer of the vaginal epithelium (P) accounted for an average of 59.1%; cells of the intermediate layer (I), 38.8%; and cells of the superficial layer (S), only 2.1% on average, ranging from 0.0 to 4.0%. Before the first treatment, out of 125 patients, none of the VMI overall scores exceeded 49%, and in all the patients, the superficial layer cells (S) percentage was less than 5%. After the first treatment, VMI did not exceed 49% in any of the patients, but only in 53.6% of the examined women the cells of the superficial layer constituted less than 5%, and in the remaining 46.4% it was ≥5%. Six weeks after the third laser treatment, in a group of 84 patients who underwent the whole therapy, the average VMI result for this group of women was 48.4% and ranged from 40.6 to 55.5. The cells of the basal layer of the vaginal epithelium (P) accounted for an average of 11.4%; the cells of the intermediate layer of the vaginal epithelium (I), 79.7%; and the cells of the superficial layer of the vaginal epithelium (S) were on average 8.9%, ranging from 7.0% to 11.0%. There was a significant increase in VMI and an increase in the number of superficial cells (S) and intermediate cells (I) and a decrease in the number of basal cells (P) compared to the state before the procedure and compared to the state after the first and second treatment. In all the patients from this group, the superficial cell count was ≥5% (Table 3, Figure 4).

The overall assessment of the quality of sexual life with the use of the female sexual function index (FSFI) of the patients before the therapy was on average 12.79 and ranged from 1.20 to 22.50. The lowest results measured in the FSFI scale were achieved in the domains of pain (1.81) and orgasm (1.91), followed by lubrication (2.03), arousal (2.30), desire (2.34), and satisfaction (2.39) (Table 4).

Six weeks after the third laser treatment, in a group of 84 patients who underwent the whole therapy, the overall result of the quality of sexual life, measured on the FSFI scale, was on average 24.39 and ranged from 20.90 to 30.30. There was a significant improvement in the quality of sexual life of patients in all domains of FSFI compared to the state before and after the first treatment. The lowest results of FSFI were achieved in the domains of pain (3.94) and desire (3.97), followed by arousal (4.02), lubrication (4.05), and orgasm (4.05), while the highest as well as after previous treatments in the domain of satisfaction (4.35). In all domains of FSFI, there was an upward trend compared to the state after the second treatment; however, the observed differences were not statistically significant (Table 5).

## 4. Discussion

During the study, a comprehensive assessment of the intensity of GSM symptoms was performed, using objective parameters (measurable parameters assessed by a doctor), such as pH, VMI, and VHIS; subjective parameters (assessed by the patients), such as FSFI; and treatment satisfaction questionnaires. Based on the available literature, it was found that studies have been performed so far that limit to assess from 1 to 3 of parameters used in this study. The most common was VHIS [1,23,24,25,26,27,28], VAS [26,27,29,30,31,32], or FSFI [28,33,34]. The VMI (VMV) analysis was much rarer [25,35].

In most studies assessing the efficiency of CO_2_ laser vaginal treatment, the intervals between particular laser sessions were 30 days apart [24,25,29,33,36,37,38] or 3–4 weeks [1,24,32,34]. In this experiment, particular laser sessions were performed in 6 weeks intervals, as was in the case of Sokol and Karram’s [6] study. This made it possible to extend the period in which the irradiated tissues were able to remodel their structure and regenerate, additionally extending the time of effective laser therapy (as a series of treatments), without increasing the number of treatments. Comparing the obtained results with the results of other authors, no significant differences were found, which proves the effectiveness of therapy conducted in this pattern.

In the conducted study, the general assessment VHIS before the treatment was on average 12.02 and ranged from 7.0 to 16.0. In the population of 43 women undergoing laser therapy by Samuels and Garcia [36], the average VHIS was 11.3 ± 3.2 (range: 5–18); in the studies of Alexiades [34]—11.83; Eder [28]—11, 93; while in the studies performed by Politano et al. [35] it was only 9.50. The study showed a significant improvement in all parameters of VHIS after the third CO_2_ laser session compared to the state before the procedure, as well as in relation to the values obtained after the first and second treatments. The overall score increased significantly, exceeding the value of 15 in the entire study population, which is the cut-off line for the diagnosis of VA [15,39,40]. After the third laser treatment, the average VHIS was 21.5 and ranged from 17.0 to 25.0. Salvatore et al. [1] obtained an average VHIS of 23.1 ± 1.9 in a group of 54 patients, while in Alexiades [34], in a population of 19 women—22.2; Pitsouni et al. [25]—20.1 ± 3.0; and Perino et al. [23]—21.5 (*p* < 0.0001). Weber et al. [41] additionally recommend a more objective assessment of GSM symptom severity by combining vaginal cytology (VMI) and pH measurement.

In the study group, the pH of the vaginal environment before the therapy was on average 5.61 and ranged from 4.7 to 7.0. “Menopausal-associated hypoestrogenism leads to physiological, histological, and anatomical changes in urogenital tissues, including thinning of the vaginal epithelium, decreased vaginal elasticity, and an increase in vaginal pH above 5” [42]. According to Simon et al. [43], pH values of 5.0–5.49 indicate mild VA, pH 5.5–6.49 moderate, and pH > 6.5 may indicate severe atrophy. Brizzolar et al. [44] consider pH > 6.0 to be incorrect and recommended as a cut-off value for VA.

The pH level of the vaginal environment decreased significantly after each session of the CO_2_ laser, both in relation to the state before the therapy and in relation to each of the preceding treatments. According to VHIS, a score of 5 is accepted for pH ≤ 4.6 [45,46], and pH > 5.0 levels are associated with a decrease in the estradiol level in patients, which indicates menopause [47,48]. GSM symptoms are associated with pH > 4.6 levels [49]. A pH value of 4.6 or higher indicates GSM, excluding patients with bacterial vaginosis [11]. In the conducted study, the lowest pH level of the vaginal environment was obtained after the third laser treatment and was on average 4.69, but in some patients was even 4.4. Before menopause, in normal conditions and without symptoms of UA, pH values are in the range of 3.6 to 4.5 [12,47]. The use of three sessions of the CO_2_ laser treatment in the conducted study allowed the patients to return to this state. In the study of Athanasiou et al. [27], in the population of 53 women (mean age 57.2 ± 5.4) after laser therapy there was a decrease in vaginal pH from an average of 5.5 ± 0.8 (initial value) to 4.7 ± 0.5 (*p* < 0.001) and improved vaginal flora (*p* < 0.001).

Before the laser treatment, the average VMI in the study group was 21.5% and ranged from 11.5 to 32.0. The cells of the parabasal layer of the vaginal epithelium (P) represent on average 59.1%, the cells of the intermediate layer of the vaginal epithelium (I)—38.8%, and the cells of the superficial layer (S) represent only 2.1% on average, ranging from 0.0 to 4.0%. Before the first treatment, no patient had a VMI greater than 49%, which is often taken as the limit of VA [26]. In all the women from the study group, the superficial layer cells represent less than 5% before the procedure. According to many authors, the cut-off value for VA is the presence of <5% of superficial layer cells [50,51,52] or the presence of >75% of parabasal layer cells [41]. 

The study showed a significant increase in the VMI after the third session of the CO_2_ laser and an increase in the number of superficial (S) and intermediate (I) layer cells, as well as a decrease in the percentage of parabasal (P) layer cells, compared to the state before the procedure and to the state after the first and second treatment. These results are confirmed in the literature [25,27,35]. In the conducted study, the VMI index was on average 48.4% and ranged from 40.6 to 55.5%. Parabasal cells of the vaginal epithelium (P) represent an average of 11.8%, cells of the intermediate layer of the vaginal epithelium (I) represent 79.7%, and cells of the superficial layer of the vaginal epithelium (S), 8.5%, ranging from 7.0 to 11.0%. In all the patients, after the third laser treatment, superficial cells represent over 5%. In studies by Pitsouni et al. [25], VMV (VMI) after the third laser treatment was on average 44.2 ± 13.7%, and superficial cells represents 3.3 ± 4.9% with the percentage of parabasal cells on average being 16.2 ± 23.9%. Politano et al. [35], 14 weeks after the third treatment, marked 5.82% P cells, 88.73% I cells, and 5.00% S cells, with the values before the therapy being 27.23%, 73.45%, and 0.50%, respectively. After three CO_2_ laser treatments, VMV (VMI) recovered to non-atrophic values in 29/55 (53%) women in the study performed by Athanasiou et al. [27].

In the conducted study, a positive correlation was observed between the assessment of VHIS and VMI, where the increase in the VHIS value was accompanied by an increase in the vaginal maturation index (VMI) and an increase in the number of superficial cells (S). Davila et al. [53] found a moderate negative correlation between the VHIS score and the VMI and showed that the severity of VA symptoms (including vaginal dryness, soreness, irritation, dyspareunia, and discharge) was not correlated with the VMI. Athanasiou et al. [27] showed after three sessions of CO_2_ laser treatment that VHIS regained the values characteristic for women without symptoms of atrophy in 44/55 (80%), while in the studies of Pitsouni et al. [25] after the third procedure, 57% (30/53) of patients achieved VMI > 49% and 89% (47/53) VHIS > 15.

The overall score of the FSFI in the studied group of patients was on average 12.79 and ranged from 1.2 to 22.5 before the therapy. The patient’s lowest score was achieved in the domains of pain (a lower score means stronger pain) and orgasm, followed by lubrication, excitement, desire, and satisfaction. In the conducted study, 6 weeks after the third treatment, a significant improvement in the quality of sexual life of patients in all FSFI domains was noted in relation to the state before and after the first treatment, and the overall result was on average 24.39, taking values in the range from 20.9 to 30.3, which proves that some patients achieved a result >27.5, which is the cut-off value. A score of ≤27.5 corresponds to clinically significant female sexual dysfunction [54,55]. Similar results were obtained in other studies by Pitsouni et al. [25], where the overall FSFI score was 25.9 ± 4.6, while Athanasiou et al. [33] after the third laser treatment obtained an average of 24.6 (15.2–35.4); Eder [28], 22.36 ± 10.40 vs. baseline value 13.78 ± 7.70; (*p* < 0.05), and Politano et al. [35], 20.55. However, in the studies of Politano et al. [35] after three laser treatments, a statistically significant improvement occurred only in the domain of desire (3.16) and lubrication (3.59), the lowest score was obtained in the domain of pain (3.18), and the highest in the domain of satisfaction (3.76), but the differences were not statistically significant compared to the pre-treatment assessment (2.55 and 3.58, respectively). In the performed study, after the third treatment, the patient’s lowest scores were achieved in the domains of pain (3.94) and desire (3.97), followed by excitement (4.02), lubrication (4.05), and orgasm (4.05), and the highest in the satisfaction domain (4.35). Obtained results prove that the CO_2_ fractional laser can improve sexual function in postmenopausal women with GSM, which is confirmed by the literature [7,33,56,57].

In the group of patients who underwent a full course of the therapy (three CO_2_ laser treatments), satisfaction levels ranged from 4.0 to 5.0, according to a 6-week follow-up evaluation after the last treatment. A total of 79.77% of patients rated their level of satisfaction as 5 (very satisfied) and 20.23% as 4 (satisfied). There were no lower scores in that group. In the studies of Salvatore et al. [1] after 12 weeks of CO_2_ fractional laser treatment, 42 women (84%) reported satisfaction with the procedure. There are studies where reported satisfaction with the procedure was achieved in 100% of patients [58].

The strength of this study was definitely the amount of objective and subjective mentors of evaluation used in the experiment. Moreover, it was a prospective design experiment with the size of the study group being 125 patients, which is one of the biggest in the available literature on this topic. A limitation of this study might be the short time of observation, however, it was one of the assumptions for this experiment.

## 5. Conclusions

Fractional CO_2_ laser therapy increases quality of life by having a beneficial effect on the sexual function of women with GSM symptoms. This positive effect was confirmed by both subjective and objective forms of GSM symptoms severity evaluation—VHIS, FSFI, or VMI. Furthermore, the obtained data suggest that it is a safe and effective method, with no significant side effects. Nevertheless, a long-term observational study with the placebo group might be a strong addition.

## Figures and Tables

**Figure 1 biomedicines-11-01304-f001:**
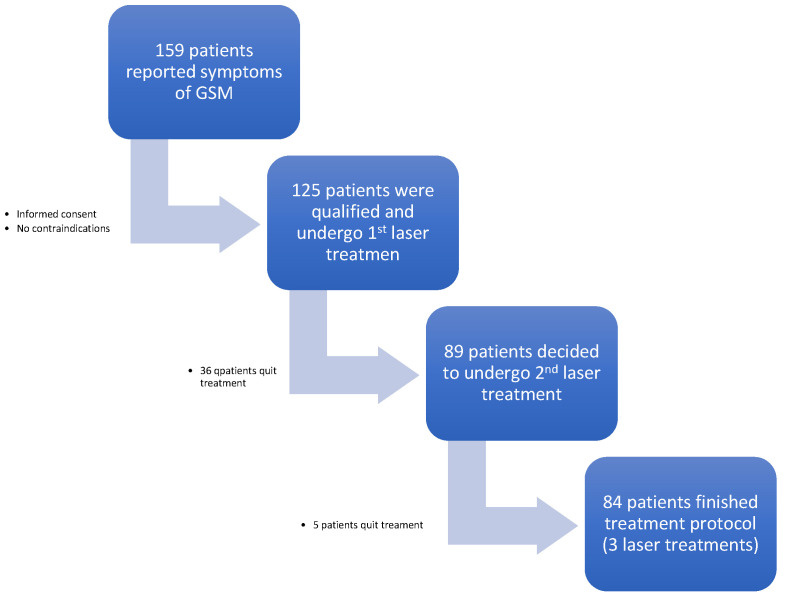
Study group.

**Figure 2 biomedicines-11-01304-f002:**
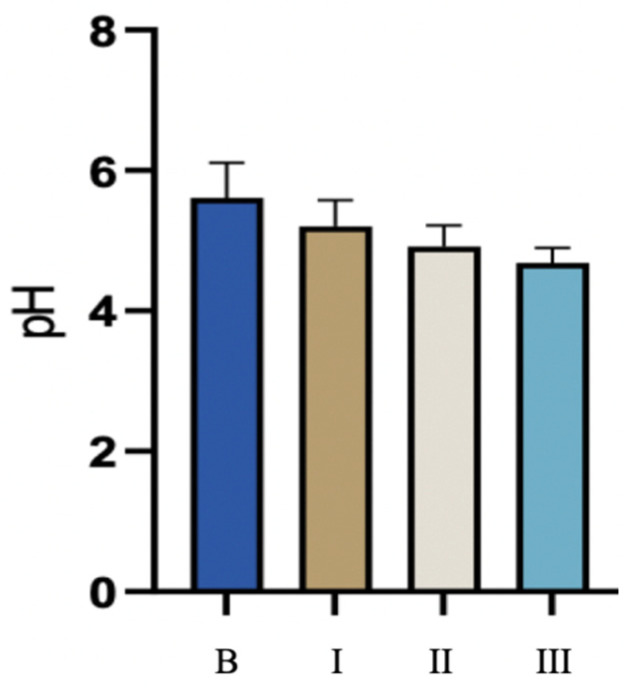
pH level before and 6 weeks after each laser treatment. B—before treatment; I—6 weeks after 1st treatment; II—6 weeks after 2nd treatment; III—6 weeks after 3rd treatment.

**Figure 3 biomedicines-11-01304-f003:**
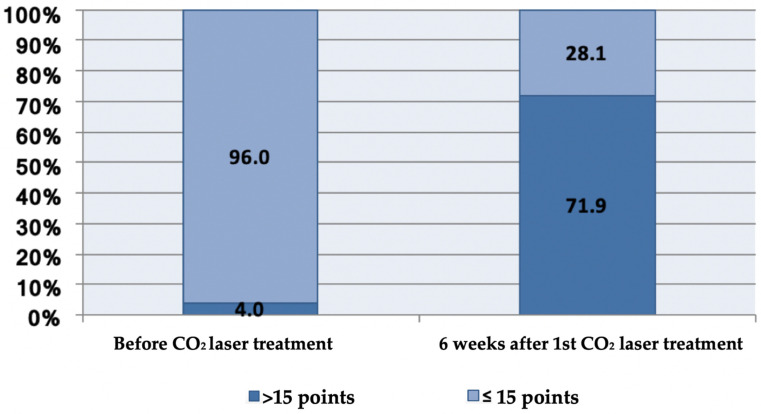
Percentage of patients (N = 125) whose overall VHIS exceeded the cut-off level for atrophy (15 points) before and after the first treatment.

**Figure 4 biomedicines-11-01304-f004:**
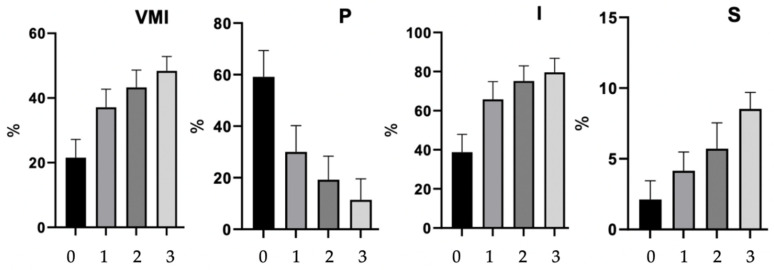
Assessment of VMI general score and particular cell percentage, after each stage of therapy. 0—before laser treatment; 1—6 weeks after 1st laser treatment; 2—6 weeks after 2nd laser treatment; 3—6 weeks after 3rd laser treatment.

**Table 1 biomedicines-11-01304-t001:** pH values of the vaginal environment during CO_2_ laser therapy.

pH	N ^1^	Mean ± SD	Median	Min	Max
Before treatment	125	5.61 ± 0.498	5.60	4.70	7.00
6 weeks after I treatment	125	5.20 ± 0.373	5.20	4.50	6.50
6 weeks after II treatment	89	4.92 ± 0.298	4.90	4.40	5.90
6 weeks after III treatment	84	4.69 ± 0.208	4.70	4.40	5.30

^1^ N—number of patients.

**Table 2 biomedicines-11-01304-t002:** Assessment of VHIS after each stage of therapy.

VHIS Factor	Before Treatment ^a^ (N= 125)	After 1st Treatment ^a^ (N = 125)	*p* ^b^	After 2nd Treatment ^a^ (N = 89)	*p* ^b^	*p* ^c^	After 3rd Treatment ^a^ (N = 84)	*p* ^b^	*p* ^c^	*p* ^d^
Elasticity	2.70	3.63	****	4.12	****	****	4.66	****	****	****
Vaginal secretion	2.17	3.25	****	3.55	****	**	3.92	****	****	***
pH	2.39	3.26	****	3.80	****	****	4.43	****	****	****
Epithelial mucous membrane	2.15	3.41	****	3.91	****	****	4.45	****	****	****
Moisture	2.60	3.38	****	3.72	****	***	4.05	****	****	**
VHIS (score)	12.02	16.92	****	19.10	****	****	21.50	****	****	****

^a^—results are presented as means; ^b^ *p*—the statistical significance of differences between the condition before and after the 1st, 2nd, and 3rd treatments; ^c^ *p*—the statistical significance of differences between the condition after the 1st treatment and the condition after the 2nd or 3rd treatment; ^d^ *p*—the statistical significance of differences between the condition after the 2nd treatment and the condition after the 3rd treatment; **—*p* < 0.01; ***—*p* < 0.001; ****—*p* < 0.0001.

**Table 3 biomedicines-11-01304-t003:** Assessment of VMI after each stage of therapy.

Cells Type (%)	Before Treatment ^a^ (N = 125)	After 1st Treatment ^a^ (N = 125)	*p* ^b^	After 2nd Treatment ^a^ (N = 89)	*p* ^b^	*p* ^c^	After 3rd Treatment ^a^ (N = 84)	*p* ^b^	*p* ^c^	*p* ^d^
VMI	21.5	37.1	****	43.3	****	****	48.4	****	****	****
Parabasal cells	59.1	30.1	****	19.1	****	****	11.4	****	****	****
Intermediate cells	38.8	65.8	****	75.2	****	****	79.7	****	****	***
Superficial cells	2.1	4.1	****	5.7	****	****	8.9	****	****	****

^a^—results are presented as means; ^b^ *p*—the statistical significance of differences between the condition before and after the 1st, 2nd, and 3rd treatments; ^c^ *p*—the statistical significance of differences between the condition after the 1st treatment and the condition after the 2nd or 3rd treatment; ^d^ *p*—the statistical significance of differences between the condition after the 2nd treatment and the condition after the 3rd treatment; ***—*p* < 0.001; ****—*p* < 0.0001.

**Table 4 biomedicines-11-01304-t004:** Assessment of the quality of sexual life of all patients measured on the FSFI scale before the therapy.

FSFI Domain	N	Mean ± SD	Median	Min	Max
Desire	125	2.34 ± 0.741	2.40	1.20	4.20
Arousal	125	2.30 ± 1.073	2.70	0.00	4.50
Lubrication	125	2.03 ± 0.960	2.40	0.00	3.30
Orgasm	125	1.91 ± 1.001	2.00	0.00	3.60
Satisfaction	125	2.39 ± 1.182	2.40	0.00	4.00
Pain	125	1.81 ± 0.879	2.40	0.00	3.60
General score	125	12.79 ± 5.351	14.20	1.20	22.50

N—number of patients.

**Table 5 biomedicines-11-01304-t005:** Assessment of the quality of sexual life after each treatment stage measured on the FSFI scale.

FSFI Domain	Before Treatment ^a^ (N = 125)	After 1st Treatment ^a^ (N = 125)	*p* ^b^	After 2nd Treatment ^a^ (N = 89)	*p* ^b^	*p* ^c^	After 3rd Treatment ^a^ (N = 84)	*p* ^b^	*p* ^c^	*p* ^d^
Desire	2.34	3.27	****	3.86	****	****	3.97	****	****	ns
Arousal	2.30	3.51	****	3.71	****	ns	4.02	****	****	ns
Lubrication	2.03	3.66	****	3.83	****	ns	4.05	****	**	ns
Orgasm	1.91	3.53	****	3.85	****	*	4.05	****	****	ns
Satisfaction	2.39	3.94	****	4.13	****	ns	4.35	****	**	ns
Pain	1.81	3.82	****	3.69	****	**	3.94	****	****	ns
General score	12.79	21.30	****	23.09	****	**	24.39	****	****	ns

^a^—results are presented as means; ^b^ *p*—the statistical significance of differences between the condition before and after the 1st, 2nd, and 3rd treatments; ^c^ *p*—the statistical significance of differences between the condition after the 1st treatment and the condition after the 2nd or 3rd treatment; ^d^ *p*—the statistical significance of differences between the condition after the 2nd treatment and the condition after the 3rd treatment; ns—no significant differences; *—*p* < 0.05; **—*p* < 0.01; ****—*p* < 0.0001.

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
