# Peer review of "Efficacy of Fractional CO2 Laser Treatment for Genitourinary Syndrome of Menopause in Short-Term Evaluation—Preliminary Study"

_biomedicines, 2023, doi:10.3390/biomedicines11051304_

Round 1

Reviewer 1 Report

I read the article with interest, however there are many criticisms.

In the abstract, the authors begin with this citation without references: "The postmenopausal state covers 40% of modern women's life and 50-70% of post-menopausal women report GSM symptoms like vaginal dryness, itching, frequent inflammations, lack of elasticity or dyspareunia. Consequently, a safe and effective method of treatment is crucial.". I STRONGLY suggest modifying the entire abstract by deleting percentages NOT referring to bibliographic indicators.

In the introduction of the papers, there is an adjective that absolutely needs to be changed: modern (in the modern world). How is it possible that the authors write such a banality?

The introduction is too long and must absolutely be shortened (mostly in the Ph and Vaginal maturation index (VMI) and laser therapy), eliminating redundancies with the discussion.

The M&Ms lack references to what type of laser treatment was performed. There are many on the market, but the equipment used must be classified.

The local ethics committee approval number must be entered.

The laser manufacturer recommends 3 treatments: specify with which modality of procedure.

Table 1 describes a drop out of patients 6 weeks after II and III treatments. It must also be specified in the text.

In addition to the studies cited (to add some recent ones not cited), the latest reviews should be included in the discussion (PMID: 35101378, 34625949, 36037664, for example).

At the end of the discussion, the following should be included: strength and limits of the manuscript.

The conclusions are somewhat trivial and must be modified, with the specificity of the study performed by the authors, namely: what this study brought new to the literature.

Author Response

Dear Reviewer 1,

In the name of my team I would like to thank you for your suggestions, which were truly helpful and accurate. The manuscript has been modified:

In abstract, there was a lack of citation to the percentage numbers – it was added.

An introduction was modified to be shorter (mostly in the Ph and Vaginal maturation index (VMI) and laser therapy), eliminating redundancies with the discussion and the phrases like “in the modern world” were deleted.

The full name, type and model of the laser used in experiment as well as modality of procedure was described in the text.

The local ethics committee approval number was entered 

According to the drop out of patients in table 1, it was also presented on the figure 1 where in methodology it was described in the text – should we also add it in the text under the table 1?

According to your suggestion the recent studies have been included in the discussion.

Strength and limits of the manuscript have been added and described.

Conclusion has been modified to be more specify, namely: what this study brought new to the literature.

I hope that the changes made, will meet your expectations. Once again, I would like to thank you for reviewing and for your suggestion to our manuscript.

Reviewer 2 Report

This study assessed the effectiveness of CO2 laser for the treatment of perimenopausal women with GSM. The methodology of this study is not clear. The inclusion criteria are not clearly stated and perimenopause is not properly defined. In addition, exclusion criteria are not stated either. Were the patients who took local oestrogen therapy excluded from the study? When did the inclusion take place? Was 125 the predefined number of participants? The study was approved by local ethics committee but no approval number is available. The evaluation of VHIS and VMI should be explained in detail in the methods section and the authors fail to do so.

Author Response

Dear Reviewer 2,

In the name of my team I would like to thank you for your suggestions, which were truly helpful and accurate. The manuscript has been modified:

Methodology was modified by stating and describing clearly the inclusion and exclusion criteria.  The inclusion took place on the first visit and primarily concerned 159 patients (predefined number of participants), where 125 patients were included in the study.

According to “perimenopause”, which was a huge linguistic mistake that we didn’t realize before, it was changed. All the patients in the experiment were postmenopausal.

According to the question: “if the patients who took local estrogen therapy were excluded from the study” the answer is: yes – it was one of the exclusion criteria.

The local ethics committee approval number was entered. 

Explanation of VMI and VHIS evaluation was added to the methodology of the manuscript.

I hope that the changes made, will meet your expectations. Once again, I would like to thank you for reviewing and the suggestions to our manuscript.

Round 2

Reviewer 1 Report

I have read the review of the paper. I congratulate the authors for the corrections reported in the text, at my suggestion. I accept the manuscript in its current form.

Reviewer 2 Report

I have no further comments